# Rewiring the Spine—Cutting-Edge Stem Cell Therapies for Spinal Cord Repair

**DOI:** 10.3390/ijms26115048

**Published:** 2025-05-23

**Authors:** Yasir Mohamed Riza, Faisal A. Alzahrani

**Affiliations:** 1Department of Biochemistry, Faculty of Science, King Abdulaziz University, Jeddah 21589, Saudi Arabia; yasirmohamedriza@gmail.com; 2Department of Biochemistry, Faculty of Science, Embryonic Stem Cell Unit, King Fahad Center for Medical Research, King Abdulaziz University, Jeddah 21589, Saudi Arabia

**Keywords:** spinal cord injury (SCI), stem cell therapy, neuroregeneration, exosome-based therapy, disability

## Abstract

Spinal cord injury (SCI) is a debilitating neurological condition that leads to severe disabilities, significantly reducing patients’ quality of life and imposing substantial societal and economic burdens. SCI involves a complex pathogenesis, including primary irreversible damage and secondary injury driven by neuroinflammation, apoptosis, and ischemia. Current treatments often provide limited efficacy, underscoring the urgent need for innovative therapeutic strategies. This paper aims to explore the potential use of stem cell (SC) therapy and exosome-based treatments as transformative approaches for managing SCI and mitigating associated disabilities. SCs, such as mesenchymal stem cells (MSCs), neural stem cells (NSCs), and embryonic stem cells (ESCs), demonstrate regenerative capabilities, including self-renewal, differentiation into neurons and glial cells, and modulation of the injury microenvironment. These properties enable SCs to reduce inflammation, inhibit apoptosis, and promote neuronal regeneration in preclinical models. Exosome-based therapies, derived from SCs, offer a novel alternative by addressing challenges like immune rejection and tumorigenicity. Exosomes deliver biomolecules, such as miRNAs, fostering anti-inflammatory, anti-apoptotic, and pro-regenerative effects. They have shown efficacy in improving motor function, reducing glial scarring, and enhancing axonal regrowth in SCI models. The objective of this paper is to provide a comprehensive review of SC therapy and exosome-based approaches, emphasizing their potential to revolutionize SCI management while addressing ethical concerns, immune rejection, and the need for large-scale clinical trials. These therapies hold promise for improving recovery outcomes and alleviating the profound disabilities associated with SCI.

## 1. Introduction

Spinal cord injury (SCI) is among the most severe neurological conditions globally, posing a significant economic burden on society due to its high disability rate [1]. While SCI is more prevalent among younger individuals, its incidence among the elderly has been steadily increasing in recent years [2]. Despite decades of research, surviving SCI patients often experience long-term, severe neurological impairments [3]. Traumatic SCI has been linked to various risk factors, including violence, extreme sports, and driving under the influence [4]. Additionally, studies indicate that over 23% of SCI patients sustain secondary injuries within 10 years, with contributing factors such as alcoholism, psychotropic drug use, and certain personality traits [5].

The pathological mechanism of spinal cord injury (SCI) can be categorized into two phases: primary and secondary injury. Primary injury occurs when the spinal cord is subjected to external forces such as contusion, tearing, compression, or vascular infarction, leading to immediate nerve damage [6]. This initial damage triggers extensive nerve cell death and disruption of the blood–spinal cord barrier. Following this, secondary injury ensues, characterized by vasospasm, hemorrhage, reactive oxygen species (ROS) formation, lipid peroxidation, inflammation, and apoptosis, which collectively exacerbate the damage through a cascade reaction [7]. Current treatment modalities for SCI, including surgery, drug therapy, hyperbaric oxygen therapy, and physical therapy, often yield unsatisfactory outcomes, necessitating the exploration of new and effective therapeutic approaches. Stem cell (SC) therapy has recently emerged as a promising avenue for SCI treatment [8].

Stem cells are unique cells capable of self-renewal, proliferation, and differentiation into various functional cell types under specific conditions [9]. SCs are classified based on their developmental stage or differentiation potential. Developmentally, they are divided into embryonic stem cells (ESCs) derived from the blastocyst and adult stem cells (ASCs), which include mesenchymal stem cells (MSCs), hematopoietic stem cells (HSCs), neural stem cells (NSCs), and induced pluripotent stem cells (iPSCs). Based on their differentiation potential, SCs are categorized as totipotent (TSCs), pluripotent (PSCs), or unipotent (USCs). SC therapy has been studied for decades, with the most notable success in hematology, where SC transplantation effectively treats various blood disorders and is now a well-established clinical practice [10]. More recently, SC therapy has been extended to neurological conditions such as intracerebral hemorrhage (ICH), ischemic stroke, traumatic brain injury (TBI), and subarachnoid hemorrhage (SAH) [11,12]. Promising findings from animal studies and clinical trials suggest that SC therapy could significantly benefit SCI patients by promoting tissue repair and nerve function recovery [11]. Commonly used SC types for SCI treatment include MSCs, HSCs, NSCs, iPSCs, and ESCs.

Research has highlighted several mechanisms through which SC therapy aids SCI recovery [13]. SC transplantation can replace or repair damaged nerve cells and tissues, including neurons and glial cells, restoring nerve conduction pathways and reconstructing function [7]. Additionally, SCs release neurotrophic factors, interact with surrounding tissues to improve the injury site microenvironment, and accelerate axonal growth. Differentiated interneurons from transplanted SCs can stimulate axon sprouting and facilitate synapse formation, connecting the spinal cord’s proximal and distal ends at the injury site [14]. SC transplantation also modulates gene expression by downregulating inflammation- and apoptosis-related genes and upregulating neuroprotective genes, thereby protecting neurons from secondary damage [15]. Some SCs differentiate into glial cells, promoting myelination and functional recovery [16].

Although progress has been made, further research is needed to optimize SC therapy for SCI. This review aims to summarize the neuroprotective mechanisms, therapeutic potential, and challenges of SC therapy, highlighting its potential use in SCI treatment in the near future.

## 2. Harnessing the Therapeutic Potential of Stem Cells for Spinal Cord Injury Recovery

Stem cell (SC) therapy holds immense promise as a treatment for spinal cord injury (SCI), utilizing a variety of SC types, each offering unique regenerative properties. Embryonic stem cells (ESCs), known for their ability to differentiate into neurons and glial cells, have demonstrated efficacy in improving motor function and alleviating neuropathic pain in animal models, with minimal adverse effects [17,18,19,20]. Mesenchymal stem cells (MSCs), derived from sources such as bone marrow (BM-MSCs), human umbilical cord (HUC-MSCs), and adipose tissue (AD-MSCs), are highly regarded for their neuroprotective capabilities. These include secreting neurotrophic factors, reducing inflammation, and promoting neural repair, with successful outcomes reported in both experimental models and clinical trials [15,21,22,23,24].

A notable Phase I clinical trial evaluated the safety of the intrathecal administration of 1 × 10^8^ autologous adipose-derived mesenchymal stem cells (AD-MSCs) in ten patients with traumatic spinal cord injury (SCI). The trial reported no serious adverse events, though non-serious events were common, and all patients successfully received the treatment. Preliminary efficacy data revealed that seven of the ten patients showed improvements in sensory and motor functions, as measured by the American Spinal Injury Association (AIS) impairment scale, with two patients advancing from AIS grade A (most severe) to grade C—exceeding typical spontaneous recovery rates. However, these results must be interpreted cautiously due to the small sample size and absence of a control group. The MRI findings indicated a benign inflammatory response, and increased vascular endothelial growth factor (VEGF) levels suggested a potential paracrine mechanism promoting tissue repair. The safety profile aligns with prior studies of AD-MSC therapy in SCI and other conditions, supporting the need for larger, randomized controlled trials to confirm its efficacy and role in SCI recovery [25].

Hematopoietic stem cells (HSCs) contribute to neurological recovery by enhancing neurotrophin expression and facilitating axon regeneration. Clinical evidence supports their safety and efficacy in restoring motor and sensory functions after SCI [26,27,28,29]. Neural stem cells (NSCs), whether endogenous or transplanted, play a vital role in reducing inflammation, repairing damaged neurons, and modulating gene expression, significantly aiding recovery [7,14,16,30]. Induced pluripotent stem cells (iPSCs) offer a versatile option due to their capacity to generate diverse cell types, but issues like low survival rates and tumorigenic risks necessitate further investigation [26,31,32,33,34,35]. A schematic of iPSC-derived organoid applications for spinal cord injury treatment is presented in Figure 1.

Other SC types, such as dental pulp stem cells (DPSCs) and olfactory ensheathing cells (OECs), have emerged as valuable candidates for SCI therapy. When used with biomaterials like 3D scaffolds, these cells enhance regeneration and functional recovery by supporting neural repair and reducing inflammation [36,37,38,39,40,41,42,43,44,45]. Collectively, these advancements underscore the transformative potential of SC therapy for SCI. However, continued research is essential to address challenges and optimize these therapies for clinical application. The characteristics of stem cell types for SCI treatment are summarized in Table 1.

## 3. Strategies for Stem Cell Therapy in Spinal Cord Injury

Stem cell (SC) therapy for spinal cord injury (SCI) employs various strategies to optimize treatment outcomes. Two key approaches include in vivo induction, where SCs are directly transplanted into the body to differentiate under local conditions, and in vitro induction, where SCs are cultured and differentiated outside the body before transplantation. Combining these techniques may enhance therapeutic effects [46,47].

SCs can be delivered through multiple pathways, including intravenous, intrathecal, intramedullary, intranasal, and intraperitoneal injections. Intravenous and intrathecal methods are less invasive, while intramedullary injections offer the precise targeting of the injury site for better cell survival and regeneration. Each route has specific advantages and challenges, necessitating further research to determine the most effective option based on the cell type and patient characteristics [48,49,50,51,52,53,54].

The number of transplanted cells is crucial for efficacy. Studies using doses from thousands to millions of SCs have shown significant therapeutic benefits, including reduced inflammation and improved motor function. However, excessively high doses may lead to reduced cell survival or adverse effects. Optimizing the dosage to balance therapeutic benefits and minimize risks remains critical [55,56,57,58,59,60,61].

Timing also plays a pivotal role. Early transplantation (within a week) can reduce secondary damage, while delayed treatment (1–3 weeks post-injury) may avoid neurotoxic environments and support better SC survival and integration. The ideal timing varies depending on the SC type and injury characteristics [62,63,64,65,66,67].

Immunosuppressive therapy is often necessary for allogeneic SC transplantation to prevent immune rejection and improve cell survival. However, nonspecific immunosuppression may cause side effects such as infections and impaired healing. Further research is needed to refine immunosuppressive protocols and understand their interactions with SCs [68,69,70,71,72,73,74,75,76,77,78,79].

By optimizing these strategies—treatment modes, delivery pathways, dosage, timing, and immunosuppression—SC therapy shows great promise for advancing SCI treatment.

## 4. Mechanisms of Stem Cell Treatment for Spinal Cord Injury

### 4.1. Inflammation Modulation

A key mechanism of stem cell (SC) therapy for spinal cord injury (SCI) is its ability to reduce inflammation [80,81]. Cheng et al. showed that transplanting neural stem cells (NSCs) into SCI rats significantly lowered neutrophil infiltration and the presence of iNOS+/mac-2+ cells at the injury site. Additionally, the mRNA levels of pro-inflammatory cytokines such as TNF-α, IL-1β, IL-6, and IL-12 were markedly reduced compared to controls, suggesting that NSCs mitigate inflammation by suppressing M1 macrophage activation and neutrophil activity, thereby enhancing neurological recovery [82].

Moreover, studies have shown that a NSC-conditioned medium can also improve neurological function by inhibiting inflammatory responses [83]. Similarly, Wang et al. reported that decellularized spinal cord scaffolds combined with bone marrow mesenchymal stem cells (BM-MSCs) decreased inflammatory cell recruitment, apoptosis, and secondary inflammation, promoting functional recovery in a spinal cord hemisection model [84].

In conclusion, SC transplantation effectively reduces inflammation, complementing other mechanisms to facilitate tissue repair and functional recovery in SCI.

### 4.2. Enhancement of Angiogenesis

Neurological recovery after spinal cord injury (SCI) depends on more than just nerve cell regeneration; it also requires the restoration of the surrounding microenvironment, including blood vessels and the extracellular matrix. Angiogenesis, the formation of new blood vessels, is a critical process that supports tissue repair and is a promising focus of therapeutic research for SCI [85].

Studies on neural stem cell (NSC) transplantation in SCI rats have demonstrated its role in promoting angiogenesis. Behavioral testing using BBB scores and an analysis of vascular endothelial growth factor (VEGF) expression through immunofluorescence and immunoblotting revealed that VEGF levels and BBB scores were significantly higher in NSC-transplanted groups compared to controls 14 days post-transplantation. These findings indicate that NSC transplantation enhances angiogenesis by stimulating VEGF expression, which contributes to improved motor function [86].

Additionally, research has shown that factors such as VEGF, angiopoietin-1, and basic fibroblast growth factor (bFGF) can stimulate angiogenesis, nerve regeneration, and neurological recovery in SCI rats. The sustained release of these angiogenic factors into the injury site significantly promoted blood vessel formation and accelerated tissue repair [87]. The extracellular matrix, as a structural component of neural tissue, plays a vital role in this process. MSC-derived components like fibronectin and cell adhesion molecules (integrin, cadherin, and selectin) further support nerve repair and axonal regeneration [88].

In summary, a combination of nutrient factors and molecular components works synergistically to reconstruct neurovascular units, ultimately enhancing neurological function.

### 4.3. Restoration and Repair

Transplanted stem cells (SCs) can differentiate into neurons and glial cells in response to the internal environment and nerve growth factors, initiating the repair and regeneration process in spinal cord injury (SCI) [89]. Studies involving the delayed transplantation of neural stem cells (NSCs) into the injured spinal cord of monkeys revealed that the transplanted cells survived and differentiated into neurons, astrocytes, and oligodendrocytes. This resulted in a reduction in lesion size and improved motor functions, including grip strength and autonomous movement, compared to the control group [90].

Zhao et al. cultured primitive NSCs derived from human embryos and transplanted them into a developing chicken central nervous system. The cells integrated into the dorsal side of the neural tube, forming clusters that differentiated into neurons. When transplanted into injured spinal cords, these primitive NSCs (originating from both NSCs and embryonic stem cells) differentiated into mature neurons and glial cells, forming functional neural circuits and promoting axonal restoration around the spinal lesion [91].

In summary, the therapeutic efficacy of SC therapy for SCI varies based on the type of transplanted cells. Further research is needed to better understand the mechanisms underlying SC-mediated spinal cord repair and to optimize these approaches.

### 4.4. Inhibition of Cell Death

Apoptosis, a key process in many neurological diseases, including spinal cord injury (SCI), is closely linked to the recovery of neurological function. A noteworthy study demonstrated that mesenchymal stem cell (MSC) transplantation in SCI rats significantly reduced the number of TUNEL-positive cells, increased neuronal survival, and improved neurological outcomes compared to untreated SCI rats [22]. Stem cell therapy was shown to downregulate proapoptotic proteins such as p53, caspase-9, caspase-3, and Bax, while upregulating antiapoptotic proteins like Bcl-2, promoting cell survival [92,93].

Further research by researchers examined various cell types, including neurons, astrocytes, macrophages/microglia, and T cells, at different time points post-transplantation. Their findings revealed that the proapoptotic factor TNF-α was significantly reduced, while the antiapoptotic factor Bcl-xL was upregulated in the SC-treated group compared to controls. These results indicate that SC transplantation can alter the balance between pro- and antiapoptotic factors as early as 1 h post-injury, reducing neuronal apoptosis and supporting the survival of tissues and motor neurons, ultimately aiding in neurological recovery [94].

In conclusion, the antiapoptotic mechanisms involved in SC therapy for SCI are of significant research interest, offering insights into improving therapeutic outcomes.

### 4.5. Nerve Growth Promotion and Regeneration

Nerve regeneration and neurotrophic support are crucial components of spinal cord injury (SCI) repair and represent significant areas of research in SCI treatment [95,96]. A note-worthy study demonstrated that adipose-derived mesenchymal stem cells (AD-MSCs) from mice could differentiate into neurogenic cells in vitro. These cells successfully survived and proliferated around the SCI lesion, as evidenced by increased levels of neurogenic markers such as Nestin, GFAP, and MAP2 [97].

Furthermore, transplanted neural stem cells (NSCs) have been shown to release neurotrophic factors to facilitate SCI repair. For example, epidermal neural crest stem cells transplanted into an in vitro SCI model, combined with valproic acid to improve the injury environment, significantly increased the expression of neurotrophic markers such as GFAP, BDNF, NT-3, and Bcl-2 seven days post-injury, highlighting the neurotrophic role of NSCs [98]. Additionally, glial cell-derived neurotrophic factor (GDNF) has been extensively studied and shown to contribute to SCI repair [99].

In summary, transplanted NSCs play a vital role in promoting nerve regeneration and providing neurotrophic support, which are essential for effective SCI recovery.

## 5. Clinical Trials and Applications of Stem Cell Therapy for Spinal Cord Injury

Stem cell (SC) therapy for spinal cord injury (SCI) has been extensively explored in clinical trials, primarily focusing on early phases (I-II) to assess safety, feasibility, and preliminary efficacy in treating neurological injuries.

Noteworthy studies have demonstrated the safety of autologous adipose-derived MSCs (AD-MSCs) in SCI patients. For instance, intravenous injections of 4 × 10^8^ AD-MSCs in eight male patients over a 3-month follow-up period revealed no serious adverse events [100]. Another trial investigated human-spinal-cord-derived NSCs in chronic T2–T12 SCI patients. Stereotactic SC injections following laminectomy and dural incision showed no severe adverse effects after 18–27 months, with some improvement in neurological function in specific spinal segments [101].

Similarly, clinical research involving autologous bone-marrow-derived MSCs (BM-MSCs) for chronic traumatic SCI demonstrated promising results. Fourteen patients received SCs injected into the lesion site post-laminectomy, with somatosensory evoked potentials (SSEPs), MRI findings, and pain scales indicating safety and potential functional improvements [102]. The intrathecal administration of BM-MSCs in patients with subacute and chronic SCI also showed no treatment-related adverse effects, even after multiple injections [103].

Additional trials have highlighted the safety and efficacy of neural stem cell (NSC) transplantation. In one study, NSC transplantation for traumatic cervical SCI resulted in no evidence of tumor formation, syringomyelia, or neurological deterioration, with notable improvements in functional scores for several patients [104]. Another investigation into intramedullary NSC injections for chronic cervical SCI reported enhanced functional outcomes and no major adverse effects over 12 months [105]. Other studies have similarly demonstrated positive outcomes, although some phase III trials, such as one using BM-MSCs, showed limited neurological improvements, with only 16 patients experiencing functional gains [23,46,106,107].

Despite encouraging findings, many clinical trials face challenges, including delays, termination, or limited patient recruitment due to funding or logistical issues. Variability in trial designs—such as differences in SCI site, severity, stage, SC type, and administration methods—makes direct comparisons difficult. To establish the true potential of SC therapy for SCI, there is a pressing need for long-term, large-scale, multicenter, and standardized randomized controlled trials. The key completed and ongoing clinical trials are outlined in Table 2.

## 6. Safety and Efficacy Considerations in Stem Cell Therapy for Spinal Cord Injury

The safety and reliability of stem cell (SC) therapy for spinal cord injury (SCI) are critical concerns. Research indicates that excessive SC doses or high infusion rates may lead to complications such as thrombosis or embolism, resulting in vascular blockage [138,139]. Additionally, transplanted SCs may trigger immune rejection, necessitating concurrent immunosuppressive therapy to mitigate this risk [66,75,140].

One of the most significant risks associated with SC therapy is tumorigenicity and cellular instability. For instance, studies observed that the DNA methylation patterns of neural stem cells (NSCs) and progenitor cells derived from induced pluripotent SCs (iPSCs) are not consistently stable, with instability increasing over cell passages [141]. Similarly, another study found that mouse bone marrow-derived MSCs (BM-MSCs) could spontaneously transform into malignant cells, forming fibrosarcoma in vivo due to chromosomal abnormalities, elevated telomerase activity, and increased c-Myc expression [142].

Other potential side effects of SC transplantation include infection, high fever, and, in severe cases, mortality [143,144]. These findings emphasize the importance of thoroughly evaluating the safety profile of SC therapies for SCI. Addressing and mitigating these risks may be even more crucial than enhancing therapeutic efficacy to ensure that SC treatments are both effective and safe for clinical use.

## 7. The Therapeutic Role of Stem Cell-Derived Exosomes in Spinal Cord Injury Recovery

Spinal cord injury (SCI) involves primary irreversible damage followed by secondary injury caused by processes like neuroinflammation, apoptosis, ischemia, and excitotoxicity [145,146]. Over the past two decades, stem cell (SC) transplantation has emerged as a promising strategy for managing central nervous system injuries by mitigating secondary damage and promoting neuronal regeneration [147]. Initially, SC therapy was thought to work primarily through differentiation into neurons and glial cells. However, recent research highlights the pivotal role of intercellular communication between transplanted SCs and surviving neural and microglial cells, facilitated largely by exosomes [148,149].

Exosomes, nano-sized vesicles containing lipids, proteins, and nucleic acids, enable paracrine interactions by delivering biomolecules like miRNAs and mRNAs to recipient cells, thereby enhancing neuroprotection and regeneration [150,151]. Exosome-based therapies circumvent challenges associated with direct SC transplantation, such as immunological rejection, tumorigenesis, and low cell viability [152]. Studies have shown that exosome administration improves motor function, reduces cavity size, enhances neuroregeneration markers, and attenuates inflammation in animal models of SCI.

In the context of spinal cord injury (SCI), exosomal microRNAs (miRNAs) have emerged as key regulators of cellular processes critical for repair and recovery. For instance, neuronal-derived exosomal miR-124-3p has been shown to suppress the activation of pro-inflammatory M1 microglia in SCI mouse models by modulating the PI3K/AKT/NF-κB signaling pathway. Additionally, studies have demonstrated that exosomal miR-494 mitigates inflammation and neuronal apoptosis in the injured spinal cord, thereby promoting nerve regeneration and enhancing motor function recovery in SCI rats. These miRNAs, sourced from cells such as mesenchymal stem cells, neural stem cells, and macrophages, regulate essential signaling pathways including NF-κB, PI3K/AKT, and ERK1/2, which in turn modulate inflammation, neurogenesis, angiogenesis, and cell survival. Such findings highlight the therapeutic potential of leveraging exosomal miRNAs to improve functional outcomes after SCI. However, further investigation is needed to fully elucidate the mechanisms by which these miRNAs operate within the injured spinal cord and to translate these insights into clinically viable treatments for SCI [153]. A schematic of exosome delivery via an intrathecal pump for spinal cord injury treatment is presented in Figure 2.

### Mechanisms of Action

Exosomes modulate neuroinflammation, apoptosis, and angiogenesis to create a favorable microenvironment for recovery:I.Neuroinflammation Control: Exosomes reduce pro-inflammatory responses by shifting macrophages towards the anti-inflammatory M2 phenotype, characterized by increased CD206 expression and reduced Iba-1 levels. Exosomes also carry miRNAs (e.g., miRNA-125a, miRNA-216a, and miRNA-23b) and siRNAs that inhibit inflammasome activation and suppress the NF-κB signaling pathway, thereby decreasing pro-inflammatory cytokines like TNF-α, IL-1β, and IL-6 [154,155,156,157,158,159,160,161,162].II.Apoptosis Inhibition: Exosome treatment upregulates anti-apoptotic proteins (e.g., Bcl-2) while downregulating pro-apoptotic factors (e.g., Bax and caspase-3). miRNAs such as miRNA-21 and miRNA-19 play a key role in regulating apoptosis-related genes, contributing to reduced neuronal death and improved recovery [163,164,165,166,167,168].III.Angiogenesis Promotion: Exosomes stimulate the formation of new blood vessels by delivering angiogenic factors, enhancing vascular stability, and mitigating ischemic damage. This mechanism is crucial for improving blood flow and supporting neuronal regeneration [148,169,170].IV.Regulation of Glial Scarring: Exosomes reduce astrocyte activation, as shown by decreased GFAP levels, and potentially shift astrocytes towards the neuroprotective A2 phenotype, which supports recovery by releasing anti-inflammatory and neurotrophic factors [169,171,172,173,174].

Recent meta-analyses confirm the therapeutic potential of exosomes in SCI recovery. For example, pooled data analyses have shown significant improvements in locomotor scores (e.g., BBB, BMS) and reductions in pro-inflammatory cytokines (e.g., IL-1β, TNF-α) following exosome treatment. Anti-apoptotic effects were evident through increased Bcl-2 and reduced Bax expression, while markers of neuronal regeneration such as GAP-43 and NeuN were significantly elevated [175,176,177].

While exosomes show promising results, their long-term safety and optimal dosage require further exploration. Studies suggest that modifying exosomes through preconditioning (e.g., hypoxic environments or inflammatory stimuli) or delivering them in fractionated doses could enhance their therapeutic efficacy [178]. These findings provide a strong foundation for designing clinical trials to validate the efficacy of exosome-based therapies for SCI. The mechanisms of action of exosomes are detailed in Table 3.

## 8. Cell-Based Treatments for Spinal Cord Injury (SCI)

Induced pluripotent stem cells (iPSCs) have become a vital asset in crafting cellular therapies for spinal cord injury (SCI). Research shows that iPSCs can transform into multiple cell types critical for SCI management, such as neural progenitor cells, oligodendrocyte progenitor cells, astrocytes, and microglia. Implanting these cells into damaged spinal cords in animal studies has yielded encouraging outcomes, with evidence of improved functionality noted in various reports [34,179]. Below, we explore the essential components of iPSC-derived cellular treatments for SCI.

### 8.1. Neural Progenitor Cells

Neural progenitor cells (NPCs) originating from iPSCs possess the ability to develop into diverse neuronal and glial cell varieties [180,181], positioning them as a promising option for replacing lost cells in SCI. When NPCs are transplanted into injured spinal cords, they contribute to positive effects like axon regrowth, myelin restoration, and synapse development [182,183]. These actions support the reconstruction of neural pathways and boost motor and sensory abilities in animal models [184]. For example, a study by Lu et al. (2012) showed that human iPSC-derived NPCs, when introduced into a primate SCI model, significantly enhanced motor capabilities, as measured by behavioral assessments [185]. Similarly, Nori et al. (2011) found that transplanting these cells into mice with SCI markedly improved motor recovery, evaluated through behavioral testing [186].

Beyond direct regeneration, NPCs also provide secondary benefits by releasing neurotrophic factors and adjusting the immune environment, aiding tissue healing and reducing inflammation in the damaged spinal cord [187]. They produce substances like brain-derived neurotrophic factor (BDNF) and glial cell-derived neurotrophic factor (GDNF) to promote neuron survival and axon growth [188], as well as anti-inflammatory agents like interleukin-10 (IL-10) to lessen local inflammation [189]. NPCs further release factors such as CNTF and NT-3 [190], supporting repair and inflammation control.

These varied attributes not only address cell loss but also enhance the environment for natural repair processes, making iPSC-derived NPCs a promising avenue for effective SCI therapies. Future efforts should refine transplantation techniques, including timing, quantity, and delivery approaches, to fully unlock their healing potential.

### 8.2. Oligodendrocyte Progenitor Cells (OPCs)

Oligodendrocytes insulate nerve fibers with myelin to support signal transmission [191,192]. SCI often disrupts this, impairing function. iPSC-derived OPCs promote remyelination and recovery [193,194]. A Phase I trial with embryonic stem cell-derived OPCs (GRNOPC1) confirmed safety and suggested tissue preservation in thoracic SCI patients [195]. Transplanted OPCs mature into oligodendrocytes, forming new myelin and boosting conduction [196]. Animal studies show motor improvements, with OPCs releasing factors like IGF-1 and enhancing repair via PDGF-AA expression, addressing both demyelination and inflammation [197,198,199].

### 8.3. Astrocytes

Astrocytes stabilize the nervous system, aiding neurons and the blood–brain barrier [200]. iPSC-derived astrocytes, when transplanted into SCI rodent models, improve neuron survival, axon regrowth, and motor function (BBB scale) [201], supporting cells in vitro and in vivo [202,203]. They reduce inflammation with IL-10 and TGF-β [204,205], secrete growth factors like CNTF and GDNF [139,206], and aid repair. However, they contribute to glial scars that hinder regrowth [207,208], which may be mitigated using chondroitinase ABC [209].

### 8.4. Microglia

Microglia, the nervous system’s immune cells, balance stability and injury response [210,211], with dual roles in SCI [212,213]. iPSC-derived microglia, though not yet widely transplanted, help study inflammation and recovery [214]. Patient-specific microglia reveal activation (e.g., TREM2 [215,216]) and neuroprotection pathways (e.g., CX3CR1/CX3CL1 [217]). This informs therapies to enhance repair and reduce harm [218,219], potentially combinable with other approaches [220].

### 8.5. Multi-Modal Therapeutic Approaches

SCI’s complexity drives combined therapies. Pairing iPSC-derived neural progenitor cells (NPCs) with scaffolds improves survival and motor recovery [221]. Integrating NPCs, OPCs, astrocytes, and microglia targets regrowth, myelination, and inflammation [222,223], enhanced by scaffolds and growth factors [224,225].

#### 8.5.1. Biomaterial Scaffolds

Scaffolds support iPSC therapy by aiding cell survival and integration [224,226]. Natural (e.g., collagen), synthetic (e.g., PLGA), or hybrid types offer unique benefits [227,228,229], providing growth cues [230,231,232]. Some, like chitosan-based ones, reduce inflammation [229], while others release factors like NGF or BDNF [233,234], promoting repair [235,236].

In addition, bioinspired hydrogel (HADA/HRR + NT3/Cur) is presented as a strategy to counteract the inhibitory fibrotic scar and hostile extracellular matrix environment after spinal cord injury. This hydrogel actively manipulates the scar tissue into a permissive, aligned substrate that guides axonal regrowth and supports neuronal relays, ultimately leading to the formation of heterogeneous, target-specific neural connections and significant functional recovery in preclinical models. This approach exemplifies how biomaterials can initiate regenerative biological processes for SCI repair [237].

Advanced biomaterials have revolutionized 3D stem cell constructs for spinal cord injury (SCI) repair by providing both structural support and the ability to influence stem cell behavior, differentiation, and integration [238,239,240,241]. Hydrogels excel in promoting efficient tissue regeneration from a single stem cell population and act as 3D carriers for mechanically stimulating encapsulated cells, enhancing cell survival and integration [242,243]. Meanwhile, poly(ethylene glycol)-fibrinogen hydrogels, initially developed to differentiate pluripotent stem cells into cardiac tissue, show promise for adaptation to neural regeneration in SCI contexts [244]. Nanomaterials, paired with stem cells, offer versatile solutions for tissue engineering, improving support and therapeutic delivery [245]. Microgels, with their shape-forming capabilities, serve as smart building blocks, enabling the construction of complex structures that replicate the spinal cord’s architecture [246]. These advancements tie into bioinspired approaches, such as the HADA/HRR + NT3/Cur hydrogel, which converts inhibitory scar tissue into a permissive environment for axonal regrowth, fostering neuronal relays and functional recovery in preclinical models. Collectively, these biomaterials highlight the need for ongoing research to refine regenerative strategies for clinical SCI applications [247].

#### 8.5.2. Growth Factors

Growth factors enhance iPSC treatments, supporting survival and repair. BDNF aids neuron growth [248], VEGF boosts blood vessels [249], and NT-3 supports development [250]. They trigger pathways like TrkB [251], TrkA [252], and FGF receptors [253], delivered via modified cells [254], scaffolds [255], or controlled systems [256].

#### 8.5.3. Electrical Stimulation

Electrical stimulation enhances iPSC therapy by boosting neural activity and recovery [257,258,259]. Combined with transplants, it improves cell integration [260], controls inflammation [261], and fosters synapses [262], aiding plasticity and function [263,264].

#### 8.5.4. Decellularized Extracellular Matrix (dECM) Scaffolds

Decellularized extracellular matrix (dECM) scaffolds, such as injectable hydrogels, electrospun scaffolds, and bioprinted scaffolds, play a vital role in spinal cord injury (SCI) treatment by offering tissue compatibility, low immunogenicity, and the ability to replicate the natural 3D structure with bioactive components [265,266,267]. Acellular spinal cord (ASC) scaffolds, frequently enhanced through modifications or composite materials for improved strength, are commonly employed to transport stem cells, drugs, or neurotrophic factors [268,269,270,271]. Furthermore, dECM scaffolds sourced from tissues like meninges and optic nerves have also been investigated for SCI applications [272,273].

## 9. Management of Chronic Spinal Cord Injury

Managing chronic spinal cord dysfunction, whether resulting from traumatic injuries or acquired myelopathies, involves addressing a wide array of persistent medical and functional challenges. The primary goal of care in this phase is to maximize functional independence and improve the patient’s quality of life [274].

This requires comprehensive strategies targeting common complications that arise long after the initial injury. These include managing cardiopulmonary and autonomic issues like autonomic dysreflexia and increased cardiovascular risks, addressing genitourinary problems such as neurogenic bladder dysfunction through techniques like intermittent catheterization and pharmacologic interventions, and managing gastrointestinal complications like constipation or incontinence with structured bowel routines [274].

Furthermore, chronic musculoskeletal issues like contractures and osteoporosis, pain (particularly challenging neuropathic pain), and spasticity are managed using various physical and pharmacological approaches. Psychosocial support is also an integral part of chronic care. Neurologic rehabilitation, focusing on functional goals and techniques like intensive locomotor training for individuals with incomplete injuries, is crucial for improving mobility and function. Future advancements in chronic SCI treatment may include neuromodulation and the further development of regenerative therapies like stem cells [274].

## 10. Stem Cell Therapy: Timing and Dosage Considerations

Optimizing stem cell (SC) therapy for spinal cord injury (SCI) involves critical decisions regarding the timing and the number of transplanted cells. The ideal time window for SC transplantation is debated, with some advocating for early intervention, such as within the first week post-injury, to potentially mitigate secondary damage and promote nerve recovery. Conversely, others suggest that delayed transplantation, weeks after the injury, might be beneficial as this would avoid the initial highly toxic microenvironment. The optimal timing appears to be influenced by factors including the specific type of SC used and the administration route [17].

Regarding dosage, most studies demonstrating significant therapeutic effects in SCI models have utilized cell numbers ranging from tens of thousands to millions. While an adequate number of cells is necessary for therapeutic benefit, and higher doses can sometimes enhance outcomes by promoting differentiation and modulating growth factors, simply increasing the cell count does not always improve survival or efficacy and can even have negative effects at very high numbers. The precise optimal number of SCs for SCI treatment remains an area requiring further research [17].

## 11. Conclusions and Future Directions

Stem cell (SC) therapy presents a promising therapeutic approach for spinal cord injury (SCI), offering significant potential for enhancing recovery through neuroprotection, neuroregeneration, and modulation of the injury environment. However, several challenges must be overcome before it can be widely applied in clinical practice. Key concerns include ethical issues, immune rejection, adverse reactions, complications, tumorigenicity, and the need for robust cell purification techniques. These limitations pose significant risks to treatment safety and efficacy, necessitating careful consideration and further investigation.

The detailed mechanisms underlying SC therapy, along with strategies to control side effects, are still not fully understood. Addressing these gaps is crucial to advancing SC-based treatments. Furthermore, the current reliance on animal models highlights the urgent need for large-scale, multicenter clinical trials to validate findings and ensure the translatability of experimental results to human patients. Such trials will provide critical insights into optimizing SC therapy and its application for SCI recovery.

Emerging approaches, such as targeting embryonic stem cells (ESCs) or combining SCs with innovative technologies such as bio scaffolds, hold promise for advancing SCI treatment. For instance, the use of mesenchymal stem cell (MSC)-derived exosomes has shown encouraging results in animal models, improving locomotion by reducing neuroinflammation, inhibiting apoptosis, and fostering neuronal regeneration. These findings underscore the potential use of cell-free exosome therapies as a safer and more efficient alternative to traditional SC transplantation.

With the rapid evolution of SC technology, breakthroughs in clinical applications are anticipated. Future research should prioritize refining therapeutic protocols, mitigating risks, and exploring novel combinations of SCs with advanced delivery systems or adjunctive therapies. As these developments unfold, SC therapy is poised to play a transformative role in the treatment of SCI, offering new hope for patients and clinicians alike.

## Figures and Tables

**Figure 1 ijms-26-05048-f001:**
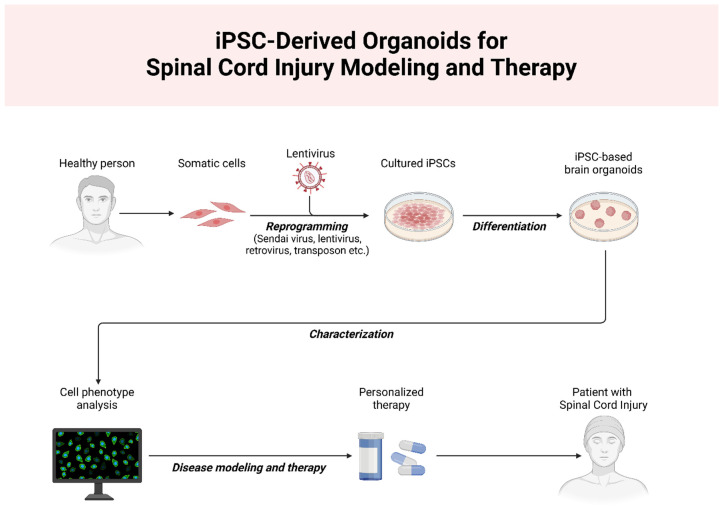
Schematic representation of the application of iPSC-derived organoids for spinal cord injury modeling and therapy. Somatic cells from healthy individuals are reprogrammed into induced pluripotent stem cells (iPSCs) using viral reprogramming techniques. These iPSCs are cultured and differentiated into specialized organoids, which can be used for cell phenotype analysis, disease modeling, and personalized therapeutic interventions. This approach holds promise for understanding disease mechanisms and advancing individualized treatment strategies for spinal cord injury patients.

**Figure 2 ijms-26-05048-f002:**
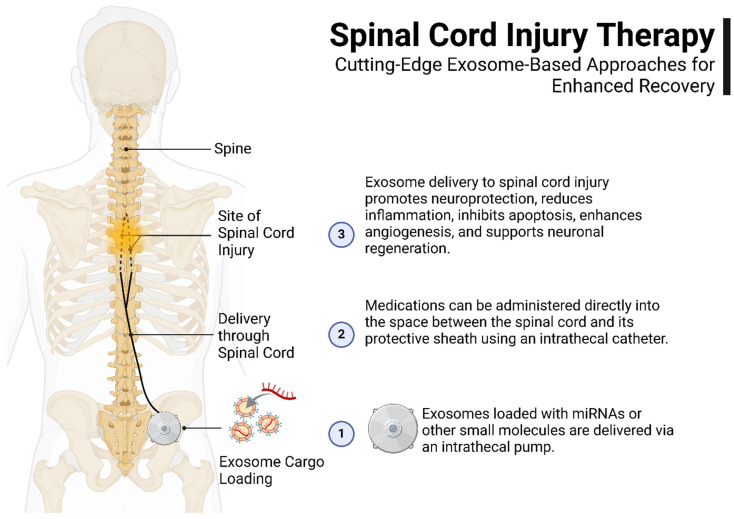
Exosomes loaded with therapeutic molecules, such as miRNAs, are delivered to the site of injury via an intrathecal pump. This approach promotes neuroprotection, reduces inflammation, inhibits apoptosis, enhances angiogenesis, and supports neuronal regeneration. Medications are administered directly into the intrathecal space, providing targeted delivery for improved recovery outcomes.

**Table 1 ijms-26-05048-t001:** Summary of stem cell types for SCI treatment.

Stem Cell Type	Key Features	Applications	References
**ESCs**	Differentiate into neurons and glial cells; Express neuron-specific antigens; Reduce neuropathic pain and improve motor function in animal models; Minimal adverse effects.	CNS disease treatment; SCI pain and function improvement.	[17,18,19,20]
**MSCs**	Includes BM-MSCs, HUC-MSCs, and AD-MSCs; Cross BBB; Reduce inflammation; Promote neuroprotection, axonal growth, and motor function recovery; Clinical and experimental evidence supports safety and efficacy.	Stroke, SCI; Reducing inflammation and apoptosis; Promoting axonal growth.	[15,21,22,23,24]
**HSCs**	Enhance neurotrophin expression; Promote oligodendrocyte and fiber formation; Improve sensory and motor functions; Safe in clinical studies.	SCI sensory and motor recovery; Safe for long-term outcomes.	[26,27,28,29]
**NSCs**	Endogenous and exogenous NSCs can repair neurons, reduce inflammation, and promote axonal regeneration; Induce anti-inflammatory responses; Improve motor recovery.	SCI repair; Reduce inflammation; Promote axonal growth and recovery.	[7,14,16,30]
**iPSCs**	Highly versatile; Differentiate into neurons and glial cells; Experimental studies show improved motor recovery; Challenges include low survival and tumorigenic risk.	Future potential in clinical application; Promising for motor recovery in SCI.	[26,31,32,33,34,35]
**Other SCs (DPSCs, OECs)**	DPSCs: Differentiate into neural-like cells; Enhance regeneration via neurotrophic factors; Supported by biomaterials like scaffolds. OECs: Modulate environment for remyelination; Reduce neuroinflammation; Clinically promising.	Promising for SCI; Enhance regeneration and reduce inflammation; Supported by biomaterials.	[36,37,38,39,40,41,42,43,44,45]

**Table 2 ijms-26-05048-t002:** Summary of key completed and ongoing clinical trials.

NCT ID	Study Title	Phase	Subjects	Cell Therapy	Route	Intervention	Efficacy	Safety	Reference	Status
**NCT02152657**	Pilot Study: Autologous MSC Transplant in Chronic SCI	1	5 (18–65 yrs)	BM-MSCs	Percutaneous	MSC Transplant	Not reported	Not reported	[108]	Completed
**NCT01325103**	Autologous BMSC Transplant in SCI Patients	1	14 (18–65 yrs)	BM-MSCs (5 × 10^6^ cells/cm^3^, single)	Intralesional	Autologous BM-MSC Transplant	Improved sensitivity, motor function, AIS, SSEP, and pain in some	One CSF leak, no severe effects	[109]	Completed
**NCT02482194**	Phase I: Autologous MSC Transplant for SCI	1	9 (18–50 yrs)	BM-MSCs (1.2 × 10^6^ cells/kg, 2–3 doses)	Intrathecal	Autologous BM-MSC Transplant	No MRI changes or ectopic tissue after 1 yr	Severe headache (1), tingling (2), no severe AE	[110]	Completed
**NCT01909154**	Safety of BMSC in Chronic Paraplegia	1	9 (18–50 yrs)	BM-MSCs (100 × 10^6^ + 30 × 10^6^ after 3 mo)	Intrathecal	Autologous BM-MSC Transplant	Sensitivity, pain, SSEP, urodynamic gains after 12 mo	AEs in all, no serious AEs	[111]	Completed
**NCT01328860**	Autologous Stem Cells for Pediatric SCI	1	10 (1–15 yrs)	BMPCs	Intravenous	Autologous BMPC Transplant	Not reported	Not reported	[112]	Completed
**NCT01186679**	Safety/Efficacy of BMSC in SCI Treatment	1/2	12 (20–55 yrs)	BM-MSCs	Intrathecal	Autologous BM-MSC Transplant	Not reported	Not reported	[113]	Completed
**NCT00816803**	Cell Transplant for SCI Patients	1/2	70 (10–36 yrs)	BM-MSCs	Intrathecal	Autologous BM-MSC Transplant	ASIA conversion (17/50), motor gains, tissue repair in some	Transient headache/pain, resolved	[114]	Completed
**NCT02570932**	Expanded BM-MSC in Chronic SCI	2	10 (18–70 yrs)	BM-MSCs (100 × 10^6^, 3 doses)	Intrathecal	Autologous BM-MSC Transplant	Not reported	Not reported	[115]	Completed
**NCT02981576**	BM-MSC vs. AT-MSC Safety/Efficacy in SCI	1/2	14 (18–70 yrs)	BM-MSCs & AT-MSCs (3 doses)	Intrathecal	Autologous BM/AT-MSC Transplant	Not reported	Not reported	[116]	Completed
**NCT01274975**	Autologous AT-MSC Transplant in SCI	1	8 (19–60 yrs)	AT-MSCs (4 × 10^8^, single)	Intravenous	Autologous AT-MSC Transplant	Reduced lesion size, ASIA A to C (1), motor/sensory gains	No severe AE, 19 AEs resolved/stabilized	[117]	Completed
**NCT01624779**	Intrathecal AT-MSC Transplant in SCI	1	15 (19–70 yrs)	AT-MSCs (9 × 10^7^/3 mL, 3 doses)	Intrathecal	Autologous AT-MSC Transplant	Not reported	Not reported	[118]	Completed
**NCT01769872**	Safety/Effect of AT-MSC in SCI	1/2	15 (19–70 yrs)	AT-MSCs (variable doses)	IV, Intrathecal, Intralesional	Autologous AT-MSC Transplant	Not reported	Not reported	[119]	Completed
**NCT01393977**	Stem Cells vs. Rehab in SCI (China)	3	34 (20–50 yrs)	UC-MSCs (4 × 10^7^)	Intrathecal	UC-MSC + Rehab Therapy	Motor, self-ability, tension gains (7/10) vs. minor rehab gains	No side effects	[120]	Completed
**NCT01873547**	Stem Cells vs. Rehab Efficacy in SCI (China)	3	300 (20–65 yrs)	UC-MSCs	Intrathecal	UC-MSC + Rehab Therapy	Not Applicable	Not Applicable	[121]	Ongoing
**NCT01321333**	HuCNS-SC in Thoracic SCI	1/2	12 (18–60 yrs)	Human CNS-SCs	Intramedullary	CNS-SC Transplant	Not reported	Not reported	[122]	Completed
**NCT01725880**	Long-Term Follow-Up of HuCNS-SC in SCI	-	12 (18–60 yrs)	Human CNS-SCs	Intramedullary	CNS-SC Transplant	Not reported	Not reported	[123]	Completed
**NCT02163876**	HuCNS-SC Transplant in Cervical SCI	2	31 (18–60 yrs)	Human CNS-SCs	Intramedullary	CNS-SC Transplant	Not reported	Not reported	[124]	Completed
**NCT02302157**	AST-OPC1 Dose Escalation in SCI	1/2	25 (18–69 yrs)	AST-OPC1 (10 M, 2 doses)	Not specified	AST-OPC1 Transplant	Not reported	Not reported	[125]	Completed
**NCT03505034**	UC-MSC in Late-Stage Chronic SCI	2	30 (18–65 yrs)	UC-MSCs (1 × 10^6^/kg, 4 doses)	Intrathecal	Allogenic UC-MSC Transplant	Not Applicable	Not Applicable	[126]	Ongoing
**NCT03521336**	UC-MSC in Sub-Acute SCI	2	84 (18–65 yrs)	UC-MSCs (1 × 10^6^/kg, 4 doses)	Intrathecal	Allogenic UC-MSC Transplant	Not Applicable	Not Applicable	[127]	Ongoing
**NCT03521323**	UC-MSC in Early Chronic SCI	2	66 (≥18 yrs)	UC-MSCs (1 × 10^6^/kg, 4 doses)	Intrathecal	Allogenic UC-MSC Transplant	Not Applicable	Not Applicable	[128]	Ongoing
**NCT03003364**	Wharton’s Jelly MSC in Chronic Traumatic SCI	1/2a	10 (18–65 yrs)	WJ-MSCs (2 doses)	Intrathecal	WJ-MSC Transplant	Not Applicable	Not Applicable	[129]	Ongoing
**NCT03308565**	AT-MSC for Traumatic SCI	1	10 (≥18 yrs)	AT-MSCs (100 M, single)	Intrathecal	Autologous AT-MSC Transplant	Not Applicable	Not Applicable	[130]	Ongoing
**NCT02574572**	BM-MSC in Cervical Chronic Complete SCI	1	10 (18–65 yrs)	BM-MSCs (2 doses)	Percutaneous	Autologous BM-MSC Transplant	Not Applicable	Not Applicable	[131]	Ongoing
**NCT02574585**	BM-MSC in Thoracolumbar Chronic Complete SCI	2	40 (18–65 yrs)	BM-MSCs (2 doses)	Percutaneous	Autologous BM-MSC Transplant	Not Applicable	Not Applicable	[132]	Ongoing
**NCT01676441**	Safety/Efficacy of BM-MSC in Chronic SCI	2/3	32 (16–65 yrs)	BM-MSCs (1 × 10^6^ & 1 × 10^7^, 2 doses)	Intrathecal	Autologous BM-MSC Transplant	Not Applicable	Not Applicable	[133]	Ongoing
**NCT02687672**	BMSC Transplant for SCI Treatment	1/2	50 (5–50 yrs)	BM-MSCs	Not specified	Autologous BM-MSC Transplant	Not Applicable	Not Applicable	[134]	Ongoing
**NCT01772810**	Safety of Spinal Cord-Derived NSC in Chronic SCI	1	8 (18–65 yrs)	Spinal Cord NSC	Not specified	NSC Transplant	Not Applicable	Not Applicable	[135]	Ongoing
**NCT04205019**	Safety of Neuro-Cells in SCI	1	10 (18–40 yrs)	Neuro-Cells	Intrathecal	Neuro-Cells Transplant	Not Applicable	No serious safety concerns or product-related adverse events	[136]	Completed
**NCT03935724**	Autologous Stem Cell Product in (Sub)Acute SCI	2/3	8 (18–65 yrs)	Neuro-Cells	Not specified	Neuro-Cells Transplant	Not Applicable	Not Applicable	[137]	Ongoing

**Table 3 ijms-26-05048-t003:** Summary of mechanisms of action of exosomes.

Summary of Mechanisms of Action of Exosomes			
Mechanism of Action			
Mechanism of Action	Function	Key Molecules/Markers	References
Neuroinflammation Control	Reduces pro-inflammatory cytokines by shifting macrophages to the anti-inflammatory M2 phenotype.	CD206, miRNA-216a, NF-κB suppression	[154,155,156,157,158,159,160,161,162]
Apoptosis Inhibition	Prevents neuronal death by modulating pro- and anti-apoptotic factors.	Bcl-2 (upregulated), Bax, caspase-3 (downregulated)	[163,164,165,166,167,168]
Angiogenesis Promotion	Stimulates blood vessel formation to enhance oxygen and nutrient delivery at the injury site.	Angiogenic factors (VEGF, bFGF)	[148,169,170]
Glial Scarring Regulation	Decreases astrocyte activation and shifts towards the neuroprotective A2 astrocyte phenotype.	GFAP (reduced), neurotrophic factors	[169,171,172,173,174]

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
