# Peer review of "Rewiring the Spine—Cutting-Edge Stem Cell Therapies for Spinal Cord Repair"

_ijms, 2025, doi:10.3390/ijms26115048_

Round 1
Reviewer 1 Report (Previous Reviewer 1)
Comments and Suggestions for Authors
Some important references are still missing. Authors need to add clinical studies with MSCs and more papers with synthetic biomaterials and decellularized matrix.
Author Response
Response to Reviewer Comments
Dear Reviewer,
Thank you for your insightful and valuable feedback. We have carefully reviewed your comment regarding the absence of references for clinical studies with mesenchymal stem cells (MSCs) and additional papers on synthetic biomaterials and decellularized matrix. To address your concerns, we have incorporated the requested references into the manuscript, with these additions highlighted in yellow for your convenience.
Clinical Studies with MSCs
We have included a notable Phase I clinical trial by Bydon et al. (2024), which evaluated the safety and preliminary efficacy of intrathecal administration of 1 × 10^8 autologous adipose-derived mesenchymal stem cells (AD-MSCs) in ten patients with traumatic spinal cord injury (SCI). This study reported no serious adverse events, with common non-serious events noted, and demonstrated that all patients successfully received the treatment. Seven of the ten patients showed improvements in sensory and motor functions, as assessed by the American Spinal Injury Association (AIS) impairment scale, with two advancing from AIS grade A to grade C—surpassing typical spontaneous recovery rates. While these findings are promising, the small sample size and lack of a control group warrant cautious interpretation. Additional observations, such as MRI evidence of a benign inflammatory response and elevated vascular endothelial growth factor (VEGF) levels, suggest a potential paracrine mechanism supporting tissue repair. This aligns with prior AD-MSC safety profiles and highlights the need for larger, randomized controlled trials to further validate efficacy.
Synthetic Biomaterials
We have expanded the discussion on synthetic biomaterials with references to advanced constructs like hydrogels, poly(ethylene glycol)-fibrinogen hydrogels, nanomaterials, and microgels (Han et al., 2025; Liao et al., 2025; Moswatsi et al., 2025; Yari-Ilkhchi et al., 2025; Nguyen et al., 2011; Wei et al., 2017; Kerscher et al., 2016; Zhao et al., 2013; Salehi et al., 2020; Zeng, 2023). These materials provide critical structural support, influence stem cell differentiation, and enhance tissue regeneration in SCI models. For instance, hydrogels promote efficient tissue regeneration and serve as 3D carriers for mechanically stimulating encapsulated cells, while poly(ethylene glycol)-fibrinogen hydrogels show adaptability for neural regeneration. Nanomaterials improve therapeutic delivery, and microgels enable complex spinal cord architecture replication. The bioinspired HADA/HRR + NT3/Cur hydrogel further exemplifies this progress by converting inhibitory scar tissue into a permissive environment for axonal regrowth and functional recovery.
Decellularized Extracellular Matrix (dECM) Scaffolds
The section on decellularized extracellular matrix (dECM) scaffolds has been enriched with references to injectable hydrogels, electrospun scaffolds, bioprinted scaffolds, and acellular spinal cord (ASC) scaffolds (Brown et al., 2022; Jiang et al., 2023; S. Liu et al., 2019; Chen et al., 2024; Liang et al., 2023; Xing et al., 2019; Yao et al., 2019; Su et al., 2023; Vishwakarma et al., 2018). These scaffolds offer tissue compatibility, low immunogenicity, and the ability to mimic the natural 3D environment, making them ideal carriers for stem cells, drugs, or neurotrophic factors. Modifications to ASC scaffolds and the use of dECM from tissues like meninges and optic nerves further enhance their utility in SCI treatment.
Conclusion
These additions provide a more comprehensive overview of the current landscape of stem cell therapy and biomaterial-based strategies for SCI repair. We believe they significantly strengthen the review and fully address your concerns, enhancing the manuscript’s scientific rigor and relevance.
Thank you once again for your constructive feedback, which has greatly improved the quality of this work.
Sincerely,
Faisal Alzahrani
Reviewer 2 Report (New Reviewer)
Comments and Suggestions for Authors
Riza and Alzahrani described in this manuscript a comprehensive review of stem cell and exosome-based therapies for spinal cord injury (SCI), with a broad spectrum from basic science to clinical application. Several critical concerns regarding the novelty, focus, and structure of the review must be addressed.
- The scope of cellular sources discussed is quite broad. The authors are encouraged to narrow their focus to three main therapeutic approaches—induced pluripotent stem cells (iPSCs), mesenchymal stem cells (MSCs), and extracellular vesicles (EVs)—to provide deeper insights and avoid the redundancy of the discussion.
- The citation papers reveal outdated literature, including references from the 1990s and early 2000s. Given the rapid advancements in exosome and cellular therapy over the past few years, this undermines the review’s relevance. Only a few references are from 2023 to 2025, which is insufficient for an overview of this area.
- The manuscript suffers from structural imbalance. Certain sections are redundant, while others are superficial.
- The manuscript lacks stratification of evidence levels (e.g., randomized controlled trials, case reports, and preclinical animal studies), which makes it difficult for readers to assess the strength of the presented claims.
Collectively, these concerns suggest that the manuscript is currently unsuitable for publication.
Author Response
Dear Reviewer,
Thank you for your feedback on our manuscript, "Rewiring the Spine—Cutting-Edge Stem Cell Therapies for Spinal Cord Repair." We value your time and effort. However, we are concerned that your comments do not reflect the significant revisions made over three rounds, which have addressed prior feedback and enhanced the manuscript’s focus, currency, and clarity. Below, we briefly address your points to clarify the manuscript’s current state.
-
Scope of Cellular Sources
You suggested focusing on induced pluripotent stem cells (iPSCs), mesenchymal stem cells (MSCs), and extracellular vesicles (EVs). Our revised manuscript already centers on these approaches, with detailed discussions in Sections 2 and 7 (e.g., MSCs in Bydon et al., 2024; iPSCs in Zeng, 2025a; EVs in Poongodi et al., 2025). Other cell types, like dental pulp stem cells, are minimally referenced for context, ensuring a streamlined and in-depth analysis. -
Currency of Literature
Your concern about outdated references is misaligned with our revisions. Over 60% of citations are from 2023–2025 (e.g., Han et al., 2025; Poongodi et al., 2025), reflecting recent advancements. Older references (e.g., Weissman et al., 2001) are limited to foundational concepts and comprise less than 10% of citations, ensuring relevance. -
Structural Balance
You noted structural imbalance, but our revised manuscript has consolidated redundant content (e.g., merging subsections in Section 4) and expanded key areas, such as clinical trials in Section 5 with Table 2. These changes ensure a balanced and logical flow, contrary to your critique. -
Stratification of Evidence Levels
You mentioned a lack of evidence stratification. However, we introduced a framework in Section 5, including Table 2, which categorizes clinical trials by phase and outcomes (e.g., Curtis et al., 2018). Sections 2 and 7 further distinguish preclinical and clinical evidence, enhancing clarity.
The manuscript now offers a comprehensive, focused, and up-to-date review, incorporating extensive revisions based on prior feedback. Unfortunately, your comments lack specific references to these changes, making it challenging to address them further. We believe the manuscript is ready for publication and would appreciate more targeted feedback if additional concerns remain.
Thank you for your consideration.
Sincerely,
Faisal A. Alzahrani
This manuscript is a resubmission of an earlier submission. The following is a list of the peer review reports and author responses from that submission.
Round 1
Reviewer 1 Report
Comments and Suggestions for Authors
This review has very broad scope. As such, many reviews were already published and therefore this review will profit from concentrating on new findings. Therefore I suggest to concentrate more on:
- Treatment with exosomes ( best part is paragraph no. 7).
- Scaffolds and their use in SCI treatment.
- Role of glial cells.
- Role of manipulation of extracellular matrix as possible method for neuroregeneration
- Data from clinical studies with appropriate citation in tables are missing.
- Concentrate more on chronic spinal cord injury treatment, since in acute SCI patients are rearly available for stem cells treatment within one week. New strategies are recently suggested which are not mentioned.
Minor comments:
References need edditing. Recent references are missing authors see 86, 90, 91, 92,100 104. REf. 101 Capitel letters? Sometimes only one author and etal.?
Figure 1: it is anout SCI not brain therefore change accordingly, or is this figure from other publication?
All tables are missing references as such it is not possible to fing the relevat data.
Author Response
-
Updating Tables with References: We have updated several tables to include references as per your suggestion, improving the access to relevant data.
-
New Section on Cell-Based Treatments: We have added a substantial new section on Cell-Based Treatments for Spinal Cord Injury (SCI), which provides a detailed discussion on combinational therapies, including bioscaffolds. This section is highlighted in yellow for your convenience.
-
Updating Figure 1: We have amended Figure 1 to better reflect the updated content based on the reviewer's feedback.
-
Discussion on Exosome-Based Therapies: We included a brief expansion on exosome-based therapies, addressing their role in spinal cord repair.
5. Splitting Clinical Trials Section: While we understand the importance of splitting the clinical trials section into ongoing and completed trials, we were unable to make this change due to time constraints and travel circumstances. We plan to address this in future revisions.
Reviewer 2 Report
Comments and Suggestions for Authors
This manuscript entitled “Rewiring the Spine—Cutting-Edge Stem Cell Therapies for Spinal Cord Repair” by Faisal and colleagues comprehensively summarized the advancement of stem cells therapy in spinal cord injury, with special emphasis on the types of stem cells involved, the underlying mechanisms, the clinical explorations, and the corresponding efficacy and challenges. This review provides a systematic perspective on the repair and treatment of spinal cord injury.
1. Multiple types of stem cells, including ESCs, MSCs, and iPSCs, have shown protective effects against SCI under different conditions, and it is therefore recommended that these be referenced in the table summaries for easy access by readers.
2. SCIs are categorized into different phases including acute, subacute, and chronic phases, and it is recommended to supplement contents regarding timing, dosage, and frequency of stem cell therapeutic interventions.
3. Stem cell therapy alone faces many shortcomings such as low survival rate and weak targeted repair ability. Emerging research has discovered stem cells based on gene editing, engineering modifications, and their integration with biomaterials, nanotechnology, etc., which greatly improves the ability of stem cells to survive, differentiate, and repair. Thus it is recommended that the author consider adding relevant content as appropriate.
4. The section describing the progress of clinical trials can be divided into completed clinical trials and ongoing clinical trials, describing their findings and safety assessments, respectively.
5. Double check grammar and typography issues.
Double check
Author Response
Response to Reviewer
-
Updating Tables with References: We have updated several tables to include references as per your suggestion, improving the access to relevant data.
-
New Section on Cell-Based Treatments: We have added a substantial new section on Cell-Based Treatments for Spinal Cord Injury (SCI), which provides a detailed discussion on combinational therapies, including bioscaffolds. This section is highlighted in yellow for your convenience.
-
Updating Figure 1: We have amended Figure 1 to better reflect the updated content based on the reviewer's feedback.
-
Discussion on Exosome-Based Therapies: We included a brief expansion on exosome-based therapies, addressing their role in spinal cord repair.
-
Splitting Clinical Trials Section: While we understand the importance of splitting the clinical trials section into ongoing and completed trials, we were unable to make this change due to time constraints and travel circumstances. We plan to address this in future revisions.
-
Adding Additional Sections or Recommended Tables: We appreciate your suggestions for additional sections or tables, and we are currently working on incorporating these. However, due to our current situation, we've focused on improvements to existing content.
-
Clarification on Phases of SCI & Emerging Research: We have included further details regarding treatment phases and ensured recent advancements in stem cell therapies are discussed in the manuscript.
-
Grammar and Typography: We have conducted a thorough check for any grammatical and typographical errors to enhance the overall readability of the manuscript